# Visible-light-switched electron transfer over single porphyrin-metal atom center for highly selective electroreduction of carbon dioxide

Deren Yang [1,4], Hongde Yu[1,4], Ting He [1], Shouwei Zuo [2], Xiaozhi Liu [3], Haozhou Yang[1], Bing Ni [1], Haoyi Li [1], Lin Gu[3], Dong Wang [1] & Xun Wang [1]

External fields are introduced to catalytic processes to improve catalytic activities. The light field effect plays an important role in electrocatalytic processes, but is not fully understood. Here we report a series of photo-coupled electrocatalysts for $CO_2$ reduction by mimicking the structure of chlorophyll. The porphyrin-Au catalyst exhibits a high turnover frequency of 37,069 $h^{-1}$ at −1.1 V and CO Faradaic efficiency (FE) of 94.2% at −0.9 V. Under visible light, the electrocatalyst reaches similar turnover frequency and FE with potential reduced by ~130 mV. Interestingly, the light-induced positive shifts of 20, 100, and 130 mV for porphyrin-Co, porphyrin-Cu, and porphyrin-Au electrocatalysts are consistent with their energy gaps of 0, 1.5, and 1.7 eV, respectively, suggesting the porphyrin not only serves as a ligand but also as a photoswitch to regulate electron transfer pathway to the metal center.

[1] Key Lab of Organic Optoelectronics and Molecular Engineering, Department of Chemistry, Tsinghua University, Beijing 100084, China. [2] Beijing Synchrotron Radiation Facility, Institute of High Energy Physics, Chinese Academy of Sciences, Beijing 100049, China. [3] Beijing National Laboratory for Condensed Matter Physics, Institute of Physics, Chinese Academy of Sciences, Beijing 100190, China. [4] These authors contributed equally: Deren Yang, Hongde Yu. Correspondence and requests for materials should be addressed to X.W. (email: wangxun@mail.tsinghua.edu.cn)

Converting $CO_2$ to high-value chemicals is an ongoing challenge in the catalytic field[1]. Thermocatalysis[2,3], photocatalysis[4–6], and electrocatalysis[7–10] are recognized as promising technologies for $CO_2$ conversion. However, these catalytic routes still suffer from a series of issues, such as high temperature for thermal catalysis, low yield for photocatalysis, and high overpotential for electrocatalysis[11–14].

In order to overcome such challenges, researchers propose that external field input can significantly improve intrinsic activity and energy efficiency. For example, a plasma field can vibrationally excite $CO_2$ molecules into free radicals and excited species, which may largely reduce activation temperature in thermocatalytic process[15]. In addition, external bias is widely used in photocatalytic processes to facilitate charge separation[16]. It is worth mentioning that suitable light irradiation possibly interferes with electronic properties of electrocatalysts, such as electron transfer, band-bending, charge distribution, Fermi level, and desorption energy of intermediate, and all these factors can alter catalytic pathways and performance. However, there have been limited studies looking into the light field effect on electrocatalytic reduction of $CO_2$. According to these limited studies, "photo-coupled electrocatalyst" is usually constituted with a ternary complex, including the dye as light harvester, semiconductor as charge transfer mediator, and metal nanoparticle as $CO_2$ activator[17,18]. On the one hand, the hybrid system easily leads to complicated synthesis and poor durability; on the other hand, the high interior resistance causes the low efficiency.

As we all know, chlorophyll continuously provides energy and resources for the entire ecosystem by the photosynthetic conversion of $CO_2$ to glucose. It is worth mentioning that porphyrin is the photosensitive core component of chlorophyll, whose porphyrinic ligand cooperates with a centrally bound Mg atom[19,20]. The electronic property is susceptible to conjugative perturbation of $18\pi$ aromatic macrocycle, thus utilizing rational external fine-turning (such as light irradiation) can drastically alter electronic performance[21]. Furthermore, extensive studies have indicated Au and single-Co-atom electrocatalysts display the best catalytic activity for CO production among metals and single-atom catalysts[22,23], respectively, and Cu species are sole candidates for C2+ production. Therefore, synthetic mimics of chlorophyll may promote insight into the catalytic mechanism and development of "photo-coupled electrocatalyst" at the atomic level via replacing the central Mg of a chlorophyll molecule with Au, Cu, or Co[24] (Fig. 1a).

Here we synthesize zirconium porphyrinic metal-organic framework (MOF) hollow nanotubes (HNTMs) as supports to anchor Au, Cu, and Co (named as HNTM-Au-SA, HNTM-Cu-SA, and HNTM-Co-SA), respectively. In the dark, HNTM-Au-SA exhibits an ultrahigh turnover frequency (TOF) of $37,069 \, h^{-1}$ at $-1.1$ V. Motivated by light irradiation, a similar TOF value is obtained at a lower overpotential with a positive shift of 130 mV. Interestingly, we observed similar results on HNTM-Cu-SA and HNTM-Co-SA, suggesting that the light irradiation can easily facilitate electrochemical activation of $CO_2$ molecule over single porphyrin-metal atom catalyst. Through both experimental tests and density functional theory (DFT) calculations, we have successfully demonstrated the feasibility of "photo-coupled electrocatalysis." Thus, the application of a "photo-coupled electrocatalyst" that integrates electrocatalytic activity with light sensitivity provides an avenue to activate $CO_2$ at a low overpotential.

## Results

### Characterization of single-Au-atom structure. As shown in Supplementary Fig. 1, HNTM could be synthesized via a solvothermal method with $ZrCl_4$, tetrakis (4-carboxyphenyl)-porphyrin and benzoic acid. Transmission electron microscopy (TEM) images reveal the uniform HNTM morphology with a size of around 500 nm (Supplementary Fig. 2a, d). The inherent hollow interior of HNTM could serve as a collector for $CO_2$ concentration and a microreactor for $CO_2$ conversion simultaneously[25].

Then the HNTM were treated with $HAuCl_4 \cdot 4H_2O$ ($CoCl_2$, $CuCl_2 \cdot 2H_2O$) in $N,N$-dimethylformamide (DMF) at 80 °C for 4 h to immobilize corresponding metal atoms (HNTM-M-SA) (Fig. 1b and Supplementary Fig. 2b, e). The HNTM loaded with metal nanoparticles (HNTM-M-NP) were also prepared for comparison (Supplementary Fig. 2c, f). The TEM and scanning tunneling electron microscopy (STEM) images exhibit that HNTM-Au-SA retains its initial nanostructure after metal atom immobilized (Supplementary Fig. 3a, b). Neither sub-nanometer clusters nor nanoparticles are detected, indicating Au species possibly exist as single atoms. The line-scanning spectra and energy dispersive spectroscopy (EDS) mapping images indicate that C, N, O, Zr, and Au elements are well-dispersed over the entire nanostructure (Fig. 1c and Supplementary Fig. 3c, d). Low loading amount of gold (0.07% by optical emission spectrometry-inductively coupled plasma (OES-ICP)) results in weak Au element signal. As shown in Fig. 1d, clearly observed porphyrin units on HNTM-Au-SA surface are marked by red rhombus. Aberration-corrected high-angle annular dark-field imaging (HAADF)-STEM images confirm that the Au species are in single atom form, which are presented as bright dots in the center of red circles (Fig. 1e). As for HNTM-Au-NP, the TEM and STEM images prove the existence of Au nanoparticles with a size distribution between 5 and 10 nm (Supplementary Fig. 4). Due to the insufficient anchoring sites, excessive addition of metal salt would result in the formation of nanoparticles.

As shown in Fig. 2a, $N_2$ adsorption–desorption isotherms of HNTM and HNTM-Au-SA exhibit similar type IV curves. The H3-type hysteresis loops indicate the existence of abundant mesopores with the sizes of 2–5 nm, which is also confirmed by pore size distribution in Supplementary Fig. 5. Compared with HNTM with the specific surface area of $894 \, m^2 \, g^{-1}$, the incorporation of Au atoms and Au nanoparticles result in lower value of 384 and $31 \, m^2 \, g^{-1}$ for HNTM-Au-SA and HNTM-Au-NP, respectively.

Both of HNTM and HNTM-Au-SA show the X-ray diffraction (XRD) patterns in consistence with PCN-225[26] (Fig. 2b), indicating the well-maintained crystallinity after single Au atom immobilized. An apparent diffraction peak located at 6–7° is observed in HNTM-Au-NP, which may be ascribed to the high degree of ordered Au plane within MOF framework[27].

X-ray photoelectron spectroscopy (XPS) was used to investigate the oxidation state of Au in HNTM-Au-SA and HNTM-Au-NP. The survey XPS spectrum reveals the predominant presence of Zr, C, O, N, and Au elements, and no other hetero-elements (Cl) are detected (Supplementary Fig. 6). The Au 4 f XPS spectrum of HNTM-Au-SA displays two peaks at binding energies (BEs) of 92.0 eV (Au $4f_{5/2}$) and 88.2 eV (Au $4f_{7/2}$), corresponding to $Au^{3+}$, which could be ascribed to Au-N coordination[28] (Fig. 2c).

To further confirm the single-Au-atom structure, X-ray absorption fine structure (XAFS) spectroscopy was conducted (Fig. 2d). The Au L3-edge in the X-ray absorption near edge structure (XANES) curve of HNTM-Au-SA is almost the same as $HAuCl_4 \cdot 4H_2O$, suggesting the oxidation state of Au atom is around +3, which is in good accordance with XPS results. Meanwhile, the XANES curves of HNTM-Au-NP and Au foil are approximately the same, implying Au species mainly exist as metallic nanoparticles.

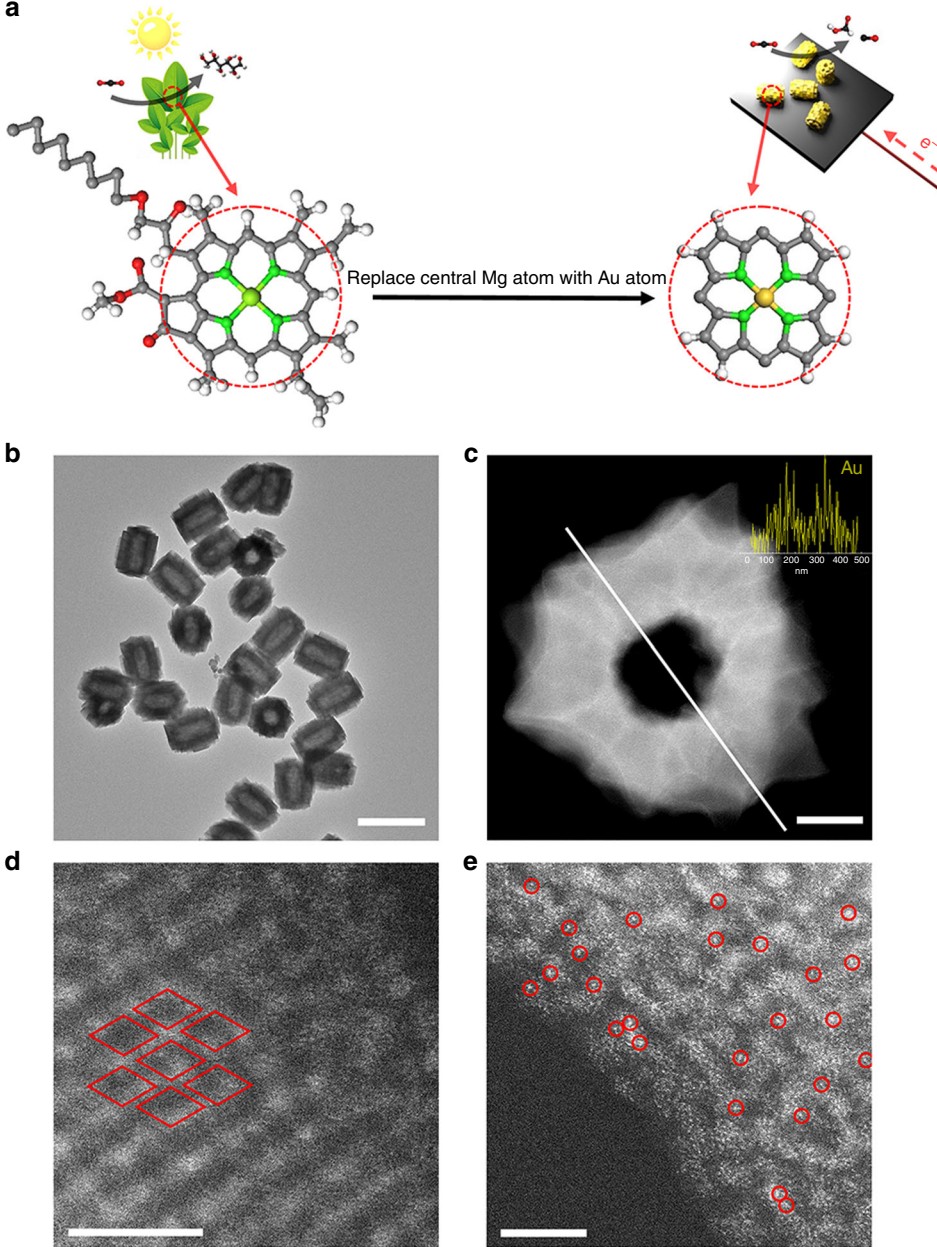

**Fig. 1** Preparation and nanostructure characterization of HNTM-Au-SA. **a** Schematic illustration of photosynthesis and photoelectrochemical reduction of $CO_2$ on chlorophyll and HNTM-Au-SA, respectively. **b** TEM image. **c** STEM image (inset is line-scanning spectrum for Au element). **d** HRTEM image. Each rhombus represents a porphyrin unit. **e** HAADF STEM image. Single Au atoms are highlighted in red circles. Scale bar: **b** is 1 μm, **c** is 50 nm, **d**, **e** is 5 nm

The coordination information of single-atom structure was indicated by the extended XAFS (EXAFS). Figure 2e shows the Fourier transform (FT) k3-weighted EXAFS spectra of HNTM-Au-SA at Au L3-edge. In contrast to reference samples (Au foil and HAuCl$_4$·4H$_2$O), no obvious peaks of Au-Au (2.1 and 2.7 Å), Au-H (0.9 Å), and Au-Cl (1.8 Å) coordination are detected. Only a prominent peak at 1.5 Å is observed, corresponding to Au-N coordination[28]. As a comparison, the EXAFS spectra of HNTM-Au-NP exhibits strong Au-Au peaks, in good agreement with XANES results. Wavelet transform was also conducted to search the atomic dispersion of Au atom. The intensity maximums of Au foil and HNTM-Au are 2.6 Å and 1.5 Å, corresponding to Au-Au and Au-N contributions, respectively[28] (Supplementary Fig. 7).

Figure 2f shows the fitting curve of single-Au-atom structure, which is perfectly reproduced by the experimental FT-EXAFS data. As shown in Supplementary Table. 1, the coordination number (Au-N) is 4, the bond length is 1.52 Å, and the disorder is 0.00785 Å$^2$. Based on the above results and previous study[24], we propose the schematic model of HNTM-Au-SA in the inset of Fig. 2f. A single Au atom is coordinated with four N atoms and anchored in the center of square-planar porphyrin unit to form a catalytic active site.

The single-atom structure of HNTM-Cu-SA and HNTM-Co-SA was also corroborated by HADDF-STEM and EXAFS analysis. As shown in Supplementary Fig. 8 and 9, no nanoparticles are detected on the MOF framework, implying the single-atom structure of Co and Cu. The aberration-corrected HAADF STEM images and EDS mapping display Cu and Co atoms are well-dispersed over the entire structure (Supplementary Fig. 8–10). XPS and EXAFS analysis further confirm their

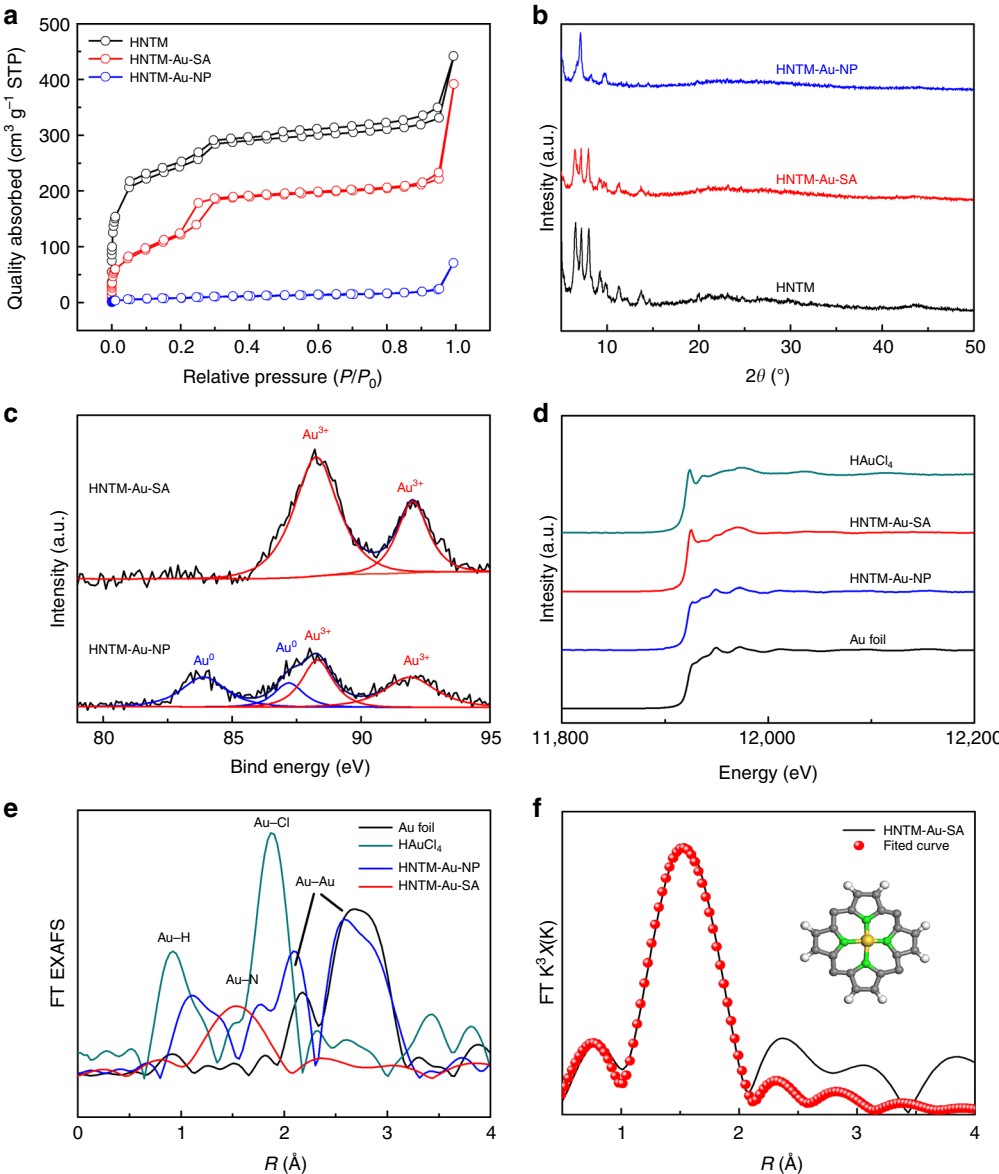

**Fig. 2** Single-Au-atom structure characterization. **a** Nitrogen adsorption–desorption isotherms and **b** XRD curves of HNTM, HNTM-Au-SA, and HNTM-Au-NP. **c** Au 4 f XPS spectra of HNTM-Au-SA and HNTM-M-NP. **d** The normalized XANES spectra. **e** FT-EXAFS spectra of HNTM-Au-SA, HNTM-Au-NP, HAuCl$_4$, and Au foil at Au L3 edge. **f** The FT-EXAFS space-fitting curve of HNTM-Au-SA. Inset shows schematic models of HNTM-Au-SA, Au (yellow), N (green), C (gray), and H (white)

single-atom structure (Supplementary Fig. 11), indicating Co and Cu atoms in the oxidation state of +2. The ICP-OES results show the single atom loadings are 0.35% and 0.59% on HNTM-Cu-SA and HNTM-Co-SA, respectively (Supplementary Table 2).

**Photo-coupled electrocatalytic performances of CO$_2$ reduction.**
To evaluate the CO$_2$ catalytic reduction performance of porphyrin-metal center structure, the photoelectrochemical reduction of CO$_2$ was carried out in a transparent H-cell equipped with a 300 W Xe lamp (with > 420 nm cutoff filter, 67% solar intensity). The distance between the light source and H-cell was about 30 cm to keep electrolyte (0.1 M KHCO$_3$ without a sacrificial regent) temperature stable at 25 °C. The gas and liquid products were analyzed by gas chromatography (GC) and $^1$H NMR, respectively. Notably, no gas and liquid product signals were detected during photocatalytic process, as the absence of sacrificial agent cannot trap the photogenerated holes for CO$_2$ activation[29,30].

Figure 3a presents the linear sweep voltammograms (LSVs) of HNTM-Au-SA scanned at 5 mV s$^{-1}$ in N$_2$- and CO$_2$-saturated 0.1 M KHCO$_3$. HNTM-Au-SA exhibits a very small cathodic current density in a N$_2$ atmosphere and only H$_2$ is detected at all potential range, conversely verifying that the products are originated from the reduction of CO$_2$ (Supplementary Fig. 12). To further verify the origin of the products, we performed isotope labeling experiments by using $^{13}$CO$_2$ as a carbon source. The generated CO and HCOOH were analyzed by GC-mass spectrometry (MS) and $^1$H NMR, respectively. As shown in Supplementary Fig. 13a, b, only the $^{13}$CO signal ($m/z = 29$) is observed in analysis of gas mixture, which is different from the $^{12}$CO ($m/z = 28$) when $^{12}$CO$_2$ was used. For liquid product, the $^1$H NMR spectrum of the electrolyte exhibits a doublet after $^{13}$CO$_2$ electrocatalysis, which is attributed to the methine proton of H$^{13}$COO$^-$ (Supplementary Fig. 13c). In contrast, H$^{12}$COO$^-$ is observed as a singlet at 8.3 p.p.m. after $^{12}$CO$_2$ electrocatalysis. These results clearly prove that both CO and HCOOH are

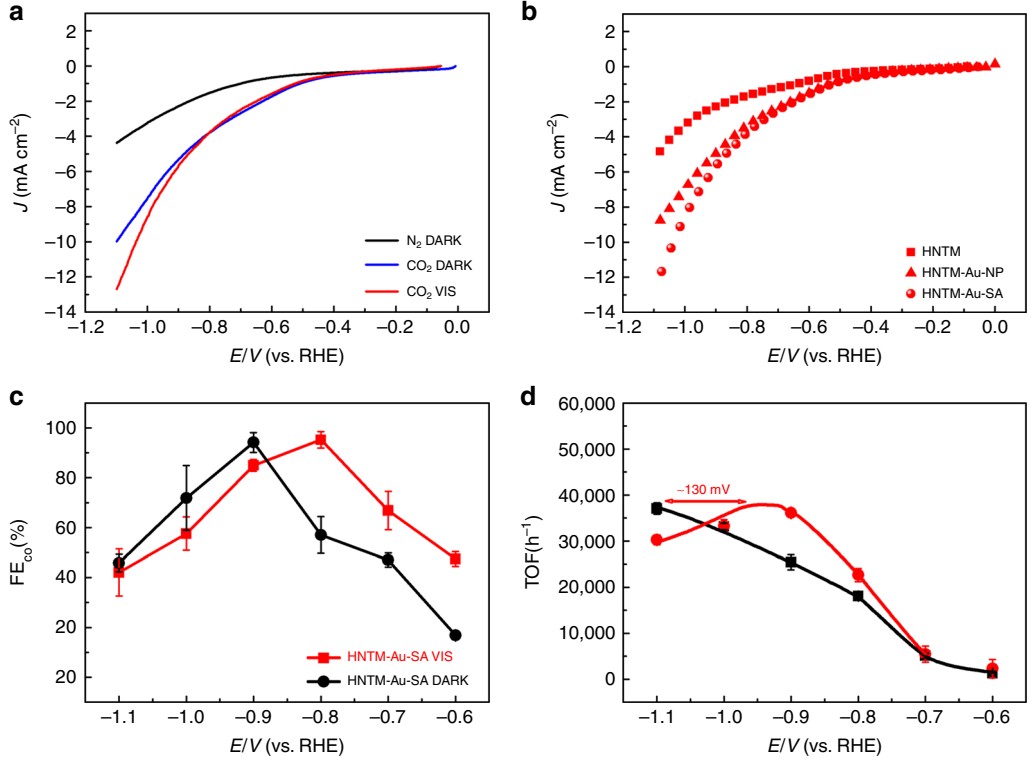

**Fig. 3** Photo-coupled electrochemical and electrochemical performance of $CO_2$ reduction. **a** LSV curves of HNTM-Au-SA scanned at 5 mv s$^{-1}$ in $N_2$-saturated (black line) and $CO_2$-saturated 0.1 M KHCO$_3$ under visible light (red line)/dark (blue line). **b** LSV curves of HNTM, HNTM-Au-SA, and HNTM-Au-NP in $CO_2$-saturated 0.1 M KHCO$_3$ under visible light. **c** FE$_{CO}$ on HNTM-Au-SA at the potentials of −0.6 V to −1.1 V under visible light (red line)/dark (black line). **d** TOF curves of HNTM-Au-SA under visible light (red line)/dark (black line). Error bars are ± s.d.

originated from $CO_2$ reduction and not from the organic residue in the electrolyte or MOF material.

With the assistance of visible light, the current density of HNTM-Au-SA significantly increases from 10 to 13 mA cm$^{-2}$ at −1.1 V in $CO_2$-saturated electrolyte (Fig. 3b). It is evident that the porphyrin can utilize solar light to transfer more electron for $CO_2$ reduction. As reference, the current densities of HNTM are negligible and almost overlapped in $N_2$- and $CO_2$-saturated electrolyte, revealing the inactivity for $CO_2$ reduction (Supplementary Fig. 14a). Furthermore, the current density on HNTM shows negligible increase under visible light, suggesting the entwined orbitals of porphyrin-Au unit can transfer more electrons from porphyrin under visible light.

The effect of visible light on Faradaic efficiency (FE) at different potentials was also investigated. As expected, the sole reaction product for HNTM is $H_2$, indicating Au is the active site for $CO_2$ reduction (Supplementary Fig. 14c). As for HNTM-Au-SA, the FE$_{CO}$ reaches a maximum of 94.2% at 0.9 V in dark condition as well as 95.2% at 0.8 V under visible light (Fig. 3c). Therefore, it is reasonable to infer that the visible light could interfere the electronic property of porphyrin to enhance atomic activity, which benefits $CO_2$ reduction occurred at a relatively low potential. It is also worth mentioning that the higher FE$_{CO}$ of HNTM-Au-SA than HNTM-Au-NP can be attributed to a more suitable chemical bond between intermediate and Au$^{3+}$ single site than Au nanoparticle (Supplementary Fig. 14d).

TOF results can help us clearly understand the light field effect in $CO_2$ activation (Fig. 3d). As expected, HNTM-Au-SA shows a maximum of 37,069 h$^{-1}$ at −1.1 V under dark condition, much higher than those previously reported electrocatalysts[31,32]. When coupled with light, a similar TOF curve of HNTM-Au-SA is obtained with a positive shift about 130 mV, indicating visible

light can easily disturb electronic property of catalysts and reduce the overpotential for $CO_2$ activation.

In order to better study the light field effect, $CO_2$ reduction performance was also recorded on HNTM-Cu-SA and HNTM-Co-SA. Similar to HNTM-Au-SA, both HNTM-Cu-SA and HNTM-Co-SA deliver larger current densities under visible light, whereas the nanoparticle counterparts perform lower activities, proving that single atom is much more active than nanoparticle due to specific valence state and maximum atom efficiency[33] (Supplementary Fig. 15). As depicted in Fig. 4a, FE$_{CO}$ of HNTM-Co-SA shows a maximum of 82.9% at −0.7 V under dark condition. Under visible light, all the FE$_{CO}$ exceed 85.0% from −0.6 to −0.8 V and a maximum of 90.4% is achieved at −0.8 V. As for HNTM-Cu-SA, only HCOOH and $H_2$ are detected with a total FE of nearly 100%. FE$_{HCOO^-}$ attains a maximum of 77.2% at −0.7 V under visible light, higher than that without light irradiation (58.3% at −0.8 V) (Fig. 4b). As expected, both HNTM-Cu-NP and HNTM-Co-NP exhibit low FE, reconfirming lower activity of metal nanoparticles (Supplementary Fig. 16). Only a weak signal of formate is detected on HNTM-Cu-NP, with a relatively low FE$_{HCOO^-}$ below 22% at all potential range.

The calculated TOF of HNTM-Cu-SA and HNTM-Co-SA at fixed potentials is plotted in Fig. 4c, d. A general trend is obtained as the peak position of TOF curves positively shifted ~ 100 mV for HNTM-Cu-SA and ~ 20 mV for HNTM-Co-SA. The shifts demonstrate once again that visible light can alter catalytic activity of different metal atoms.

The observed product and corresponding FE maximum on each catalyst are compared in Fig. 5a. HNTM-M-SA always show the highest FE$_{max}$ under visible light. To further evaluate the photo-coupled system, the mass-specific, area-specific, and charge-specific rate of CO and HCOO$^-$ on each catalyst are presented in Supplementary Fig. 17–19. All of HNTM-M-SA

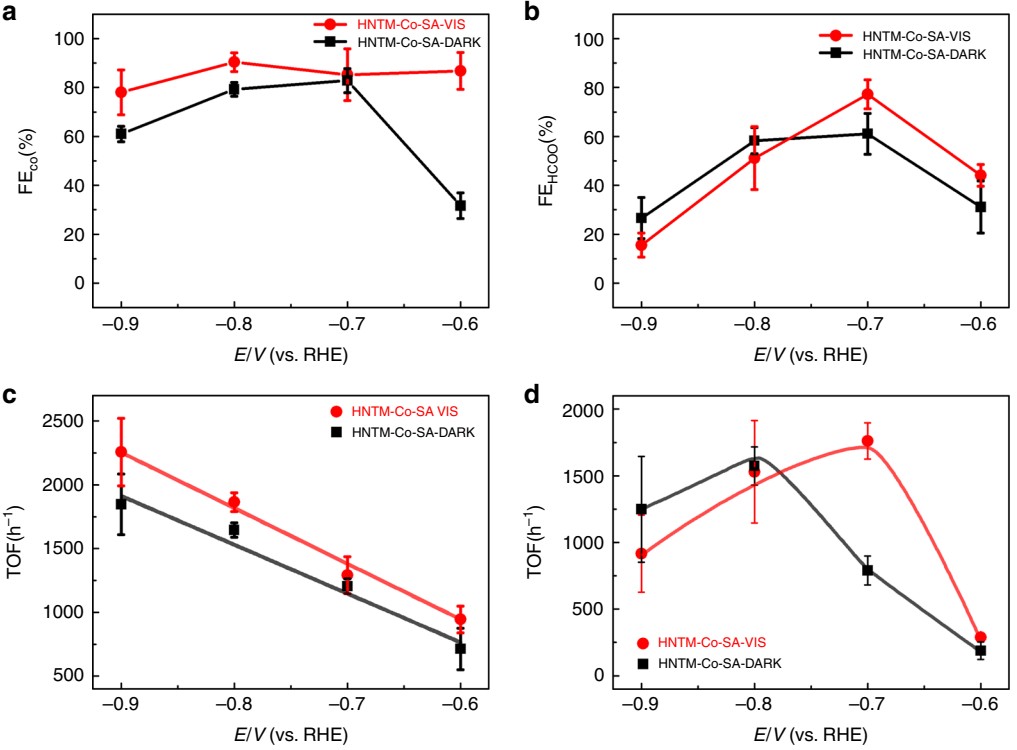

**Fig. 4** Photo-coupled electrochemical and electrochemical reduction of $CO_2$. **a** $FE_{CO}$ on HNTM-Co-SA under visible light (red line)/dark (black line). **b** $FE_{HCOO^-}$ on HNTM-Cu-SA under visible light (red line)/dark (black line). **c** TOF curves of HNTM-Co-SA under visible light (red line)/dark (black line). **d** TOF curves of HNTM-Cu-SA under visible light (red line)/dark (black line). Error bars are ± s.d.

exhibit significant improvement under visible light, verifying the light field effect. Furthermore, HNTM-Cu-SA and HNTM-Co-SA show similar performance with reported porphyrin complex; however, TOF value of HNTM-Au-SA shows at least one order of magnitude larger than them (Supplementary Table. 3). Reported photoelectrocatalysts are also listed, while they usually show complicated structures with low FE and reaction rate, which limit their practical application. Thereby, how to design and synthesize photo-coupled electrocatalysts with high performance remains the central challenge in this field.

**Photo-coupled electrocatalytic mechanism**. To understand the observed high selectivity and activity on the porphyrin-based catalysts, we performed DFT calculations of the free energy landscape for the CO pathway of $CO_2$ reduction with computational hydrogen electrode model[34]. The conversion of $CO_2$ to CO is an overall two-electron two-proton process. As demonstrated by the energy profile (Fig. 5b), the rate-determining step in this process is the first proton-coupled electron-transfer step in which the adsorbed *$CO_2$ is transformed to *COOH, and the corresponding free energy changes are 1.2 eV, 0.6 eV, and 2.8 eV, respectively, on HNTM-Au-SA, HNTM-Co-SA, and HNTM-Cu-SA catalysts, which is qualitatively consistent with previous computational studies of CO2RR on Co and Cu phthalocyanine and porphyrin monolayers, despite differences in model designs and material structures[35–38]. The energy required for the $CO_2$-to-COOH conversion on HNTM-Cu-SA is so high that it is not an active catalyst for CO production. It is further noted that different from the regular 5-coordination single-atom catalysts with only one active site[23], the metal atom in HNTM-M-SA is anchored in the porphyrin plane and coordinated by four N atoms, so that it may simultaneously adsorb two $CO_2$ molecules and catalyze the reduction reaction on both sides of the porphyrin plane. As

illuminated in Fig. 5b, $\Delta G$ of the rate-limiting step is reduced from 1.2 eV to 0.8 eV on HNTM-Au-SA when one of the adsorbed *$CO_2$ has been reduced to *COOH, whereas it is increased from 0.6 eV to 0.8 eV on HNTM-Co-SA, suggesting the $CO_2$ activation on both sides of porphyrin plane may be synergic for HNTM-Au-SA. This synergistic catalysis mechanism for HNTM-Au-SA is illustrated in Fig. 5c, in which each Au atom has two active sites and it adsorbs two $CO_2$ molecules and catalyzes their reduction synergistically. The proposed mechanism for the CO formation on HNTM-Au-SA gives rise to the lower potential of −0.9 V in corroboration with the experimental observation and the improved efficiency of $CO_2$ reduction compared with HNTM-Co-SA with non-synergistic mechanism. It has thus explained the much higher TOF observed for CO production on HNTM-Au-SA than HNTM-Co-SA (25,425 $h^{-1}$ vs. 1847 $h^{-1}$ at −0.9 V under dark).

UV-vis diffuse reflection was performed to better understand the optical properties. All samples show strong absorption in the range of 200–800 nm, which is ascribed to photon absorption ability of porphyrin[39]. After single Au atom immobilized, HNTM-Au-SA exhibits higher visible light absorption (Supplementary Fig. 20a). Furthermore, the HNMT-Au-SA shows a lowest photoluminescence (PL) intensity, suggesting the most efficient charge separation occurred on coordinated single atom than encapsulating nanoparticles (Supplementary Fig. 20b). To compare the photo-induced electron transfer efficiency, the difference value between photocurrent respond at −0.8 and 0 V was calculated. HNTM-Au-SA shows the highest value of 0.50 mA, more than five times higher than HNTM-Au-NP, confirming that visible light can accelerate electron transfer to single Au atom than nanoparticles (Fig. 6a). More importantly, Fig. 6b clearly reveals that HNTM-Co-SA shows a higher Tafel slope of 1.21 V $dec^{-1}$ under dark at the overpotential rang of 0.6–0.8 V, whereas it decreases to 0.78 V $dec^{-1}$ under visible light, indicating visible

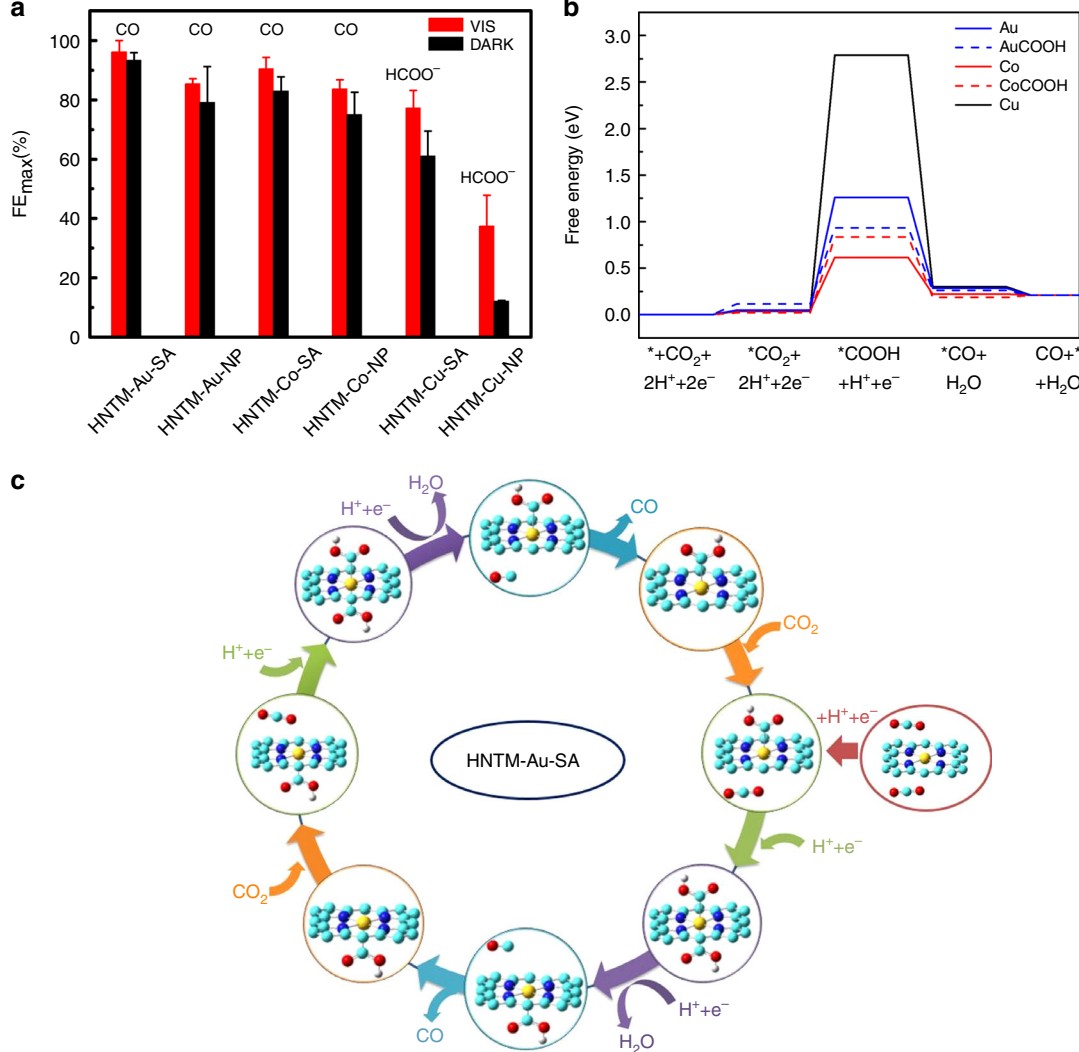

**Fig. 5** Density functional theory calculations. **a** FE maximum and product observed on each catalyst. Red and black color bars represent FE under light/dark conditions, respectively. Error bars are ± s.d. **b** Calculated free energy profile of electrochemical reduction of $CO_2$ to CO. **c** The synergistic catalysis mechanism of $CO_2$ reduction on HNTM-Au-SA

light can facilitate a faster $1e^-$ transfer from $CO_2$ to $CO_2^{\bullet-}$. We also measured the electrochemical impedance spectroscopy (EIS) to investigate the effect on the charge transfer resistance ($R$ct). As presented in Fig. 6c, EIS results show the $R$ct is 7.8 ohms under visible light vs. 8.5 ohms under dark condition, indicating the photoexcited porphyrin can efficiently facilitate charge transfer.

As illuminated by the frontier orbital analysis of porphyrin-Au, the electron of the vacant highest occupied molecular orbital (HOMO) is mostly localized on the porphyrin ligand, and that of the lowest unoccupied molecular orbital (LUMO) is mainly localized on the central metal atom. Under dark condition, external electrons can only flow to the LUMO, which need a high overpotential to excite across the HOMO-LUMO gap (in the right part of Fig. 6d). To illustrate the mechanism of photoelectrocatalysis and the effect of visible light, we further calculated the electronic properties of $S_1$ and $T_1$ excited states of these porphyrin-based catalysts. Both $S_1$ and $T_1$ exhibit clearly the ligand-to-metal charge-transfer excitation characteristic, which apparently facilitates the electron-flow and $CO_2$ activation occurring on the single-Au-site (Fig. 6e). Upon the light absorption, porphyrin is first excited from the $S_0$ ground state to the $S_1$ state, and then undergoes fast relaxations to the ground state as well as the $T_1$ triplet state through the intersystem

crossing process. The spin-orbital coupling in heavy metal elements is so strong that the relaxation to this long-lived $T_1$ state after the visible light irradiation is prominent. As all the photophysical processes occur much faster than the electrocatalysis process, we expect the photo-coupled electrocatalysis of $CO_2$ proceeds mainly from the $T_1$ state of porphyrin-Au instead of the $S_0$ state, with the same mechanism proposed above (Fig. 5c). In addition, the energy required for $T_1$ state is taken from the light absorption and no electricity is needed (Fig. 6f). The energy of $T_1$ is 1.7 eV higher than the $S_0$ state, which may result in the reduction of overpotential by ~ 130 mV (Fig. 6e). To our delight, the $T_1$ states of HNTM-Cu-SA and HNTM-Co-SA are 1.5 eV and 0 eV, respectively, consistent with the overpotential shift of ~ 100 mV and 20 mV observed in experiment.

## Discussion

In summary, we have successfully synthesized a series of photo-coupled electrocatalysts (HNTM-Au-SA, HNTM-Cu-SA, and HNTM-Co-SA) based on the structure of chlorophyll. When coupled with light, HNTM-Au-SA and HNTM-Co-SA achieve high $FE_{CO}$ of 95.2% and 92.6%, as well as $FE_{HCOO^-}$ of 77.2% on HNTM-Cu-SA. HNTM-Au-SA presents a volcano TOF curve

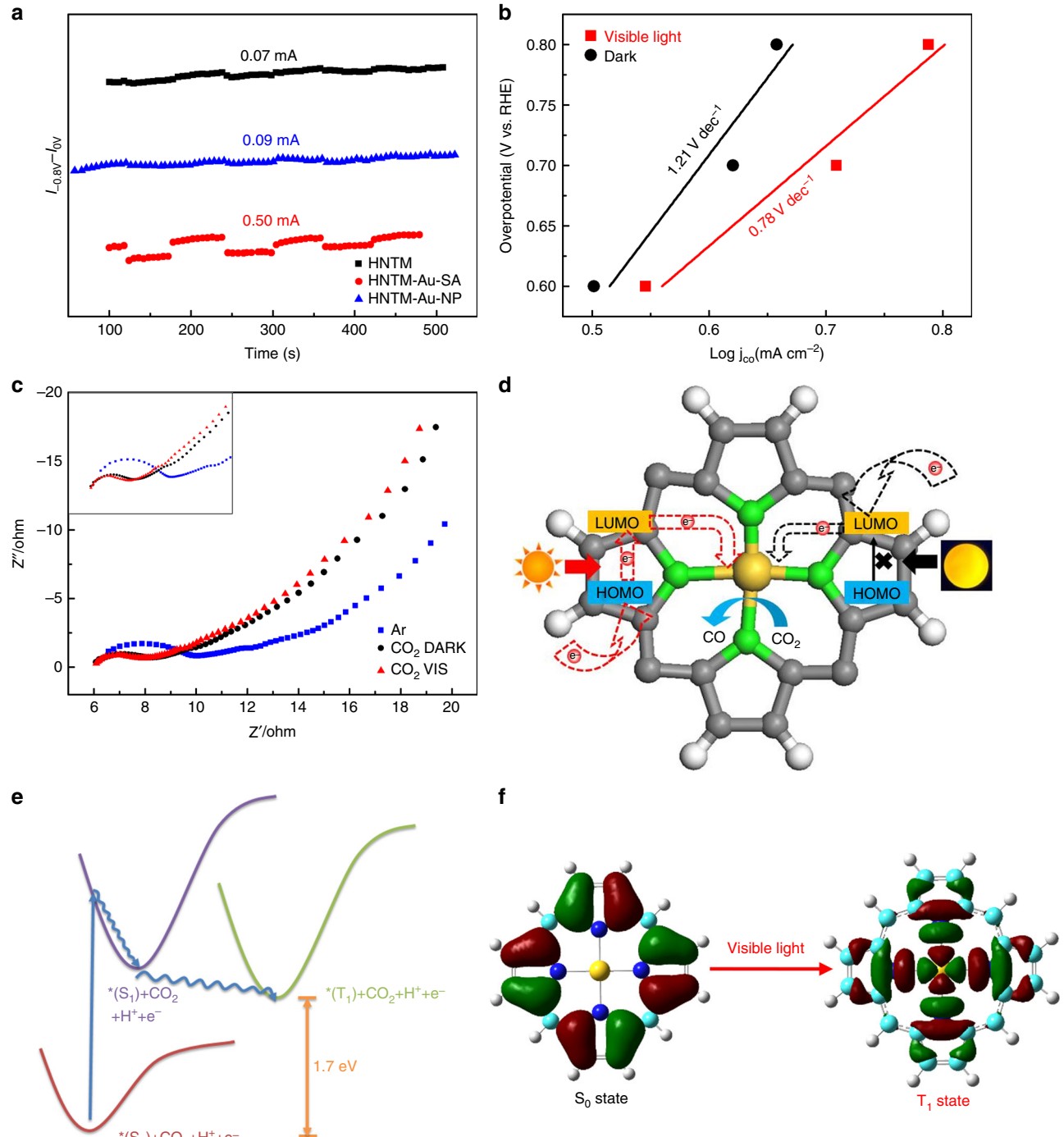

**Fig. 6** Mechanism for photo-coupled electrochemical and electrochemical $CO_2$ reduction. **a** The difference between photocurrent response at $-0.8$ V and 0 V on HNTM-Au-SA, HNTM-Au-NP, and HNTM. **b** Tafel plots of HNTM-Co-SA under visible light (red line)/dark (black line). **c** EIS spectra of HNTM-Co-SA in Ar-(blue line)/$CO_2$-saturated 0.1 M $KHCO_3$ under visible light (red line)/dark (black line). **d** The electron transfer pathways under visible light and dark are shown in the left and right parts, respectively. **e** Transition orbital of $T_1$ state under visible light. **f** Schematic representation of the excited states of porphyrin-Au

with a maximum of 37,069 h$^{-1}$ at $-1.1$ V under dark condition, whereas it positively shifts ~ 130 mV under visible light. The light-assisted TOF results show a positive shift of 20, 100, and 130 mV for HNTM-Co-SA, HNTM-Cu-SA, and HNTM-Au-SA, respectively, consistent with the calculated energy gaps of 0, 1.5, and 1.7 eV. Coupled with light field, the porphyrin ligand serves as a photoswitch to collect photons and motivate external electron transfer from ground state to $T_1$ state. On the contrary, electrons can only transfer from the high-energy LUMO under

dark conditions, resulting in high overpotential. This work provides a blueprint for the design of photo-coupled electrocatalysts at the atomic scale, utilizing porphyrin ligand as photosensitizer and coordinated metal atom as catalytic site.

## Methods
**Synthesis of HNTM-M-SA (Au, Co, Cu).** In a typical procedure, 2 mL DMF, 10 mg $ZrCl_4$, 250 mg benzoic acid, and 200 μL $H_2O$ were added to 10 mL capacity Teflon-lined autoclave in sequence. After stirring for 5 min, 10 mg TCCP (detail

synthetic process is shown Supplementary Methods) was further added in the above-mentioned solution and stirred for 10 min at room temperature. The autoclave was sealed and then heated at 120 °C for 18 h. The products were collected via centrifugation at 6000 r.p.m. for 3 min and further washed with ethanol for three times. HNTM was obtained after drying in vacuum drying oven at room temperature overnight.

In a typical procedure, 4 mL DMF, 20 mg HNTM, and 200 μL HAuCl₄·4H₂O aqueous solution (20 mg Au per mL) were added to a 10 mL Teflon-lined autoclave and then stirred for 10 min at room temperature. Next, the mixture was heated at 80 °C for 4 h. After cooling down, HNTM-Au-SA was separated via centrifugation at 10,000 r.p.m. for 3 min and further washed with ethanol for three times. For HNTM-Co-SA and HNTM-Cu-SA, the same concentration of CoCl₂ and CuCl₂·2H₂O replaced HAuCl₄·4H₂O, respectively. HNTM-M-NP was also obtained by adding doubling dose of metal salts.

**Material characterization.** The as-prepared samples were characterized by TEM (HITACHI H-7700), high-resolution TEM (HRTEM, FEI Tecnai G2 F20), XRD (Bruker D8-advance), and XPS (PHI Quantera SXM), respectively. The BEs were corrected by the C1s peak at 284.8 eV. All the single atom catalysts were performed on HAADF-STEM (Titan 80–300). Metal loading was measured by ICP-OES (IRIS Intrepid II XSP, ThermoFisher). UV-vis diffuse reflectance spectra were obtained using a UV-vis spectrophotometer (UV-3600, Shimadzu). The PL spectra were carried out on a Varian Cary Eclipse Fluorescence spectrophotometer in a range of 420–850 nm. The CO and HCOOH produced from the ¹³CO₂ labeling experiments were analyzed by GC-mass spectrometer (GC/MS-QP2010, Shimadzu) and ¹H NMR, respectively.

The XAFS spectra were measured at 1W1B station in Beijing Synchrotron Radiation Facility. XAFS measurements at the Au L3-edge, Co K-edge, and Cu K-edge were conducted in fluorescence mode using a Lytle detector. The energy was calibrated using corresponding metal foil as references. The acquired EXAFS data were processed according to the standard procedures using the Athena program.

**Photoelectrochemical measurements.** Electrochemical measurements were carried out in a transparent H-cell with an electrochemical station (CHI 660E). Photoelectrochemical measurements were performed in the above-mentioned electrochemical system equipped with a 300 W Xe lamp (with > 420 nm cutoff filter, 67% solar intensity). The distance between the light source and H-cell was about 30 cm to keep electrolyte (0.1 M KHCO₃ without a sacrificial regent) temperature stable at 25 °C. Five milligrams of catalyst and 5 mg Vulcan XC-72 were dispersed in 2 ml water–isopropanol solution with a volume ratio of 1:1. Nafion (20 μL; 5 wt%) was further added in the mixture and kept sonicating for 30 min to form a homogeneous ink. Then, 400 μL link was uniformly dropped onto the gas diffusion electrode (area: 1 cm × 2 cm) at room temperature, giving a catalyst loading of 0.5 mg cm⁻². The gas diffusion electrode, Pt wire, and Ag/AgCl were used as working electrode, the counter electrode, and reference electrode, respectively. Reversible hydrogen electrode (RHE) potentials were calculated by the Nernst equation, $E_{RHE} = E_{Ag/AgCl} + 0.197 V + 0.0591 \times pH$.

Before measurements, CO₂ gas (99.99%) was purged into the cathodic electrolyte at a rate of 20 s.c.c.m. for 30 min. During electrochemical and photoelectrochemical process, the flow rate of CO₂ was controlled at 7 s.c.c.m. The LSVs scanned at 5 mv s⁻¹ in N₂- and CO₂-saturated 0.1 M KHCO₃. Collected gas products and liquid products were qualitatively and quantitatively analyzed by GC and ¹H NMR. The mole number of gas products and liquid products were calculated from GC peak areas and ¹H NMR peak areas based on standard curves of pure samples, respectively. The FE was calculated according to the following equation (1):

$$FE(\%) = \frac{96485(C/mol) \times n(mol/ml) \times 2 \times 7(ml/min) \times 60(min)}{Q} \times 100\% \quad (1)$$

Where 96,485 is Faraday constant (C/mol), $n$ is the amount of products per milliliter (mol/ml), 2 is the electron transfer number (CO, HCOOH, and H₂), 7 is the flow rate of CO₂ (mL min⁻¹), 60 is reaction time (min), and $Q$ is total charge obtained from chronoamperometry.

The TOF value of catalysts was calculated by the equation (2):

$$TOF(h^{-1}) = \frac{I_{product}/2F}{mw/M} \quad (2)$$

Where TOF is TOF (h⁻¹), $I_{product}$ is the partial current density of products (A), 2 is the electron transfer number for CO and HCOOH production, $F$ is faraday constant (96,485 C mol⁻¹), $m$ is catalyst mass in the electrode (g), $w$ is metal loading (Au, Co, Cu) on HNTM, and $M$ is atomic mass.

**DFT calculations.** All the calculations including structural optimization, vibrational, and thermochemistry analysis were carried out using DFT with the Gaussian 16 package[40]. The PBE0 functional was used to treat the exchange-correlation energy and the long-range dispersion interaction was taken into account by Grimme's D3 parameters. The 6–31 G* basis set was applied for the light elements including H, C, N, and O, and the LANL2DZ basis set and pseudopotential were used for the heavy metal elements such as Co, Cu, and Au. All the metal porphyrin complex structures were first fully optimized and then fixed with only adsorbed species relaxed in the later computations. The free energies of reactants, CO₂ and H₂, were corrected to reproduce $\Delta G_{298} = 0.208$ eV for CO2RR to produce CO: $CO_2(g) + H_2(g) \rightarrow CO(g) + H_2O(l)$.

## Data availability
The data that support the plots within this paper and other findings of this study are available from the corresponding author upon reasonable request. The source data underlying Figs. 2–4, 6a–c and Supplementary Figs 5–6, 11–19 are provided as a Source Data file at https://doi.org/10.6084/m9.figshare.8845793.v1 (https://doi.org/10.6084/m9.figshare.8845793)[41].

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

## Acknowledgements

This work was supported by National Key R & D Program of China (2017YFA0700101 and 2016YFA0202801) and NSFC (21431003). We thank Beijing Synchrotron Radiation Facility (BSRF) for providing the EXAFS tests in 1W1B station. We thank Dr. Haifang Li for providing GC-MS tests.

## Author contributions

X.W. led the whole project. D.Y. synthesized the catalysts, performed TEM, XRD, XPS, XAFS, ICP, UV-vis, PL, and electrochemical and photoelectrochemical measures, data analysis, and wrote the manuscript. H.Yu performed DFT calculation and edited the manuscript. T.H. guided synthetic process. S.Z. performed XAFS measurements and analysis. X.L. and L.G. performed HAADF-STEM measurements. B.N. guided the usage of GC and performed HRTEM measurements. H.Yang and H.L. edited the manuscript. D.W. guided DFT calculation and edited the manuscript.

## Additional information

**Competing interests:** The authors declare no competing interests.

