## [Peer Review File · Nature Communications]

Reviewers' comments:

Reviewer #1 (Remarks to the Author):

The photoelectrochemical conversion of CO₂ is a very interesting research topic that deserves attention. This work presents the synthesis, characterization and use of various single-atom photoelectrocatalysts based on the structure of chlorophyll. The system is able to achieve a FE of 98.5% for CO (Au-based material), FE of 99.9% for CO (Co) and FE of 81.6% for HCOOH (for Cu), with a maximum TOF of 45067 h⁻¹ at -1 V under dark conditions, that show a positive shifts as high as 140 mV for Au-based materials. The operation principles of the analysis are clear and the manuscript is written in a very comprehensive manner. Technically speaking, the results are innovative and will be of help for researchers in the field. Overall, I believe this report may be suitable for Nat. Commun. after considering the following points:

- The rates for CO and HCOOH formation are not shown. These values may give valuable information to evaluate the performance of the system. Could the authors present formation rates normalized by catalyst loading, available reaction area and charge passed through the system? Could the authors show the experimental error in the obtained results?
- Even if the literature may offer results for different systems (electrolytes, cell configuration, conditions, etc.) it would be nice to compare the performance of the developed photoelectrocatalysts with other materials reported in literature. A brief discussion on this might assist the field in developing better photoactive catalysts.

Reviewer #2 (Remarks to the Author):

This manuscript reports the use of MOF-based systems as photoelectrocatalysts for CO₂ reduction. The main claims of the work appear to be that: (1) light allows to enhance the performance of the materials due to the specific single atom catalysis design of the system and (2) the reported performance of the Au-based system is better than other electrocatalyst reported so far.

Overall, claim (1) seems to be mostly addressed but the presentation of the data does not make justice to the work. In addition, some clarification of the results obtained (or absence of) should be addressed. Some comments on that aspect are provided below. Finally, the authors should comment on reproducibility of their results. Unless I have missed it, this seems completely absent and given the field of study, it becomes very important.

Regarding, claim (2), this should be further supported by highlighting direct comparison with catalysts tested so far. A table or graph could help in this regard.

Below are further major comments - mostly related to further strengthening claim (1):

The authors keep referring to "light field effect". What does it mean?

The first sentence of the manuscript is a bit simplistic. It sounds like utilising CO₂ as a feedstock will systematically lower CO₂ emissions but this is not necessarily the case. I suggest deleting such misleading statement.

The authors should clarify the exact hypothesis and objective of their work in the introduction. At this point, I understand that they wish to highlight and understand the role of light in the CO₂ electroreduction process but this is not so clear straight away.

The choice of Au, Co and Cu metal should be explained in the introduction, especially since the authors refer to the Mg metal in chlorophyll.

What is the metal loading for all samples (SA and NP)? Why does the loading vary so much from metal to metal? How can this allow direct comparison of the samples?

Can the authors comment/explain the absence of any photocatalytic activity?

Can the authors verify the origin of the products formed, especially since organic solvent was used during the MOF synthesis?

The authors should consider include a figure in the main text that highlight the products observed for each catalyst and the selectivity.

Other minor comments:

First sentence of abstract unclear. Maybe it can be split into two sentences.

Surface areas should not be given with decimal numbers.

How do the authors explain the absence of surface area reduction after metal loading?

"A sharp diffraction peak at 7.1° is observed in HNTM-Au-NP, corresponding to Au crystal on MOF framework." A reference is needed.

Reviewer #3 (Remarks to the Author):

The authors performed an experimental study on Visible-light-switched electron transfer over single porphyrin-metal atom center for the highly selective electrochemical reduction of CO₂

after carefully go through the whole manuscript, some comments for this work are as follows:

(1) there are some literature reported electroreduction of CO₂ via porphyrin complex. it will be better if the authors can make a comparison (in a Table) of the performance (e.g., overpotential, faradaic efficiency, TOF, durability, stability, etc) between the current system and those from literature for the electroreduction of CO₂.

This will highlight the present study and also provide useful information for the community of CO₂ reduction, and further promote the progress of this field.

(2) in the first paragraph of introduction section, e.g., on line 32, electrocatalysis of CO₂, the authors should include and cite some latest references on CO₂ reduction and single atom catalysis.

Theor. Chem. Acc. 2018. 137. 98.

ACS Sustainable Chem. Eng. 2018, 6, 15494–15502.

J. Mater. Chem. A, 2019, 7, 3805–3814

J. Mater. Chem. A, 2019, DOI: 10.1039/C9TA01188A

<https://pubs.rsc.org/en/content/articlelanding/2019/TA/C9TA01188A#!divAbstract>

it will be better if the authors can compare the present study with that from J. Mater. Chem. A, 2019, DOI: 10.1039/C9TA01188A

<https://pubs.rsc.org/en/content/articlelanding/2019/TA/C9TA01188A#!divAbstract>

this is the latest study on electroreduction of CO₂ by two dimensional poly-porphyrin monolayers. the unit cell is very close to the current porphyrin complex.

(3) on line 190, page 8, the authors used computational hydrogen electrode model (CHE) to study the pathway of CO₂ reduction. but on line 301, page 13, the authors used Gaussian 16 with PBE0+D3 functional to do calculations.

it should be pointed out that CHE model was proposed and used in the periodic systems, e.g., metal surfaces.

but Gaussian code usually used for cluster systems. The authors need to explain the more details about the calculations of CO₂ reduction and how to used CHE model with Gaussian code.

(4) there are some problems in the reference citation

ref.22 < Adv. Mater.> should be < Adv. Mater.>

ref.2 < Chem. Soc. Rev. 40, 3703-3727 (2011)> should be < Chem. Soc. Rev. 40, 3703-3727 (2011)>

there should be only a space between the journal name and the number of volume.

ref.1 please double check the name of the journal <Earth System Science Data>, make sure it is abbreviation.

After the authors correct all the above problems with a minor revision, this paper can be published after review the revision of manuscript.

Reviewer #1:**Comments:**

The photoelectrochemical conversion of CO₂ is a very interesting research topic that deserves attention. This work presents the synthesis, characterization and use of various single-atom photoelectrocatalysts based on the structure of chlorophyll. The system is able to achieve a FE of 98.5% for CO (Au-based material), FE of 99.9% for CO (Co) and FE of 81.6% for HCOOH (for Cu), with a maximum TOF of 45067 h⁻¹ at -1 V under dark conditions, that show a positive shifts as high as 140 mV for Au-based materials. The operation principles of the analysis are clear and the manuscript is written in a very comprehensive manner. Technically speaking, the results are innovative and will be of help for researchers in the field. Overall, I believe this report may be suitable for Nat. Commun. after considering the following points:

Reply: Thank you for recognizing the novelty of our work. We appreciate this reviewer's heart-felt comments and suggestions on our manuscript and we are absolutely delighted to accept the suggestions of the reviewer.

1. The rates for CO and HCOOH formation are not shown. These values may give valuable information to evaluate the performance of the system. Could the authors present formation rates normalized by catalyst loading, available reaction area and charge passed through the system? Could the authors show the experimental error in the obtained results?

Reply: According to the reviewer's suggestion, we have added the CO and HCOOH formation rate on different catalysts (Figure S16-18). The normalized results are also discussed in our revised manuscript (Page 9, Line 16-18). All of HNTM-M-SA exhibit significant improvement when coupled with light, verifying the light field effect. In addition, error bars are added in the obtained results.

2. Even if the literature may offer results for different systems (electrolytes, cell configuration, conditions, etc.) it would be nice to compare the performance of the developed photoelectrocatalysts with other materials reported in literature. A brief discussion on this might assist the field in developing better photoactive catalysts.

Reply: Thanks for your good suggestion. According to the reviewer's suggestion, we have added the performance comparison (Supplementary Table. 3) and the related discussion (Page 9, Line 21-22) in our revised manuscript.

Reviewer #2:

Comments:

This manuscript reports the use of MOF-based systems as photoelectrocatalysts for CO₂ reduction. The main claims of the work appear to be that: (1) light allows to enhance the performance of the materials due to the specific single atom catalysis design of the system and (2) the reported performance of the Au-based system is better than other electrocatalyst reported so far.

Overall, claim (1) seems to be mostly addressed but the presentation of the data does not make justice to the work. In addition, some clarification of the results obtained (or absence of) should be addressed. Some comments on that aspect are provided below. Finally, the authors should comment on reproducibility of their results. Unless I have missed it, this seems completely absent and given the field of study, it becomes very important.

Regarding, claim (2), this should be further supported by highlighting direct comparison with catalysts tested so far. A table or graph could help in this regard.

Below are further major comments - mostly related to further strengthening claim (1):

Reply: We appreciate this reviewer's heart-felt comments and suggestions on our manuscript and we are absolutely delighted to accept the suggestions. According to the reviewer's suggestion on claim (1), experiments were performed at least for 3 times and results are shown as mean±standard deviation. Furthermore, we have compared the

performance with other catalysts in Supplementary Table. 3 and discussed it (Page 9, Line 19 to 22) in our revised manuscript.

1. The authors keep referring to “light filed effect”. What does it mean?

Reply: This is an excellent question! According to our best knowledge, “light filed effect” means that the suitable light irradiation can interfere electronic property of specific catalysts, such as electron transfer, band-bending, Fermi level and desorption energy of intermediate, and all of these factors can alter catalytic pathway and performance distinctly. To better understand it, we have added the related specification (Page 2, Line 14-17).

2. The first sentence of the manuscript is a bit simplistic. It sounds like utilizing CO₂ as a feedstock will systematically lower CO₂ emissions but this is not necessarily the case. I suggest deleting such misleading statement.

Reply: Thank you for pointing this out. We have deleted the misleading statement according to the reviewer’s suggestion in our revised manuscript.

3. The authors should clarify the exact hypothesis and objective of their work in the introduction. At this point, I understand that they wish to highlight and understand the role of light in the CO₂ electroreduction process but this is not so clear straight away.

Reply: Thanks for your good suggestion. We have clarified and highlight our hypothesis and objective in the introduction (see Page 2, Line 7-13).

4. The choice of Au, Co and Cu metal should be explained in the introduction, especially since the authors refer to the Mg metal in chlorophyll.

Reply: Extensive studies have indicated Au and single-Co-atom catalyst displayed the most excellent electrocatalytic ability for CO production among metals (**J. Am. Chem. Soc.** 2014, 136, 40, 14107-14113) and single-atom catalysts (**Angew. Chem. Int. Ed.** 2018, 57, 7, 1944-1948; **J. Am. Chem. Soc.** 2018, 140, 12, 4218-4221), respectively. Furthermore, Cu species is the sole candidate for C₂+

production. According to the reviewer's suggestion, we have added our explanation in the revised manuscript (see Page 3, line 9-11).

5. What is the metal loading for all samples (SA and NP)? Why does the loading vary so much from metal to metal? How can this allow direct comparison of the samples?

Reply: Thank you for your professional advice. We have added the metal loading in Supplementary Table 2.

Compared with Cu (+2) and Co (+2), Au atom shows the valence state of +3 on HNTM-Au-SA (see Figure. 2C), which may be more difficult to anchor in the center of porphyrin unit, and result in lower loading amount. The low Au loading can be also confirmed by EDS elemental mapping of HNTM-Au-SA (see Supplementary Figure 3d), which shows weak Au element signal. However, HNTM-Cu-SA and HNTM-Co-SA displays strong Cu and Co signal, respectively, corresponding to higher metal loading (see Supplementary Figure 8-9).

Although the metal loadings for SA samples are different, TOF results can be fairly compared by normalizing the amount of metal atoms because every metal atom acts as an isolated catalytic site. Furthermore, in order to fairly compare the activity between single atom and nanoparticle samples, we removed TOF curves of NP samples from Figure. 3d and 4c,d and use their area-specific activity, mass-specific activity and charge-specific activity for comparison (see Supplementary Figure 16-18).

6. Can the authors comment/explain the absence of any photocatalytic activity?

Reply: According to our knowledge, porphyrin-based photocatalysts usually need sacrificial agent (such as TEA and TEOA) to trap hole to boost charge separation. In our work, although the samples are excited under light irradiation, the photogenerated electron-hole will quickly recombine in the absence of sacrificial agent, result in no photocatalytic activity. We have added related description in our revised manuscript (see Page 7, line 12-13).

7. Can the authors verify the origin of the products formed, especially since organic solvent was used during the MOF synthesis?

Reply: This is an excellent question! In order to confirm the carbon source of CO₂ reduction process, control experiment in N₂ atmosphere was conducted (Supplementary Fig. 12). Only H₂ was detected at the potential of -0.6 to -1.0 V vs.RHE. It conversely demonstrates that the products are originated from the reduction of CO₂, and not from the organic residue in the electrolyte or MOF material. We have added related description (see Page 7, line 16-18).

8. The authors should consider include a figure in the main text that highlight the products observed for each catalyst and the selectivity.

Reply: We have highlighted products and selectivity for each catalyst in Figure 4e according to the reviewer's suggestion.

9. First sentence of abstract unclear. Maybe it can be split into two sentences.

Reply: We have modified the first sentence of abstract in the revised manuscript according to the reviewer's suggestion (see Page 1, line 3-5).

10. Surface areas should not be given with decimal numbers.

Reply: We have corrected this mistake in the revised manuscript. Thank you!

11. How do the authors explain the absence of surface area reduction after metal loading?

Reply: Thank you for pointing this out! To confirm whether the surface area is changed or not, BET measurement was repeated (see Figure. 2A and Supplementary Figure 5). The specific surface area of HNTM, HNTM-Au-SA and HNTM-Au-NP is 894, 384 and 31 m² g⁻¹, respectively, far from the initial results (61, 69 and 62 m²

g^{-1}). In order to protect their nanostructure, we did not adequately dry and disperse samples before the first measurement, result in much lower surface area.

In addition, the updated results indicate that the incorporation of SA and NP into HNTM can remarkably low the specific surface area, in accord with reviewer's hypothesis. The related descriptions are also corrected (see Page 5, line 5-7). Thank you!

12. "A sharp diffraction peak at 7.1° is observed in HNTM-Au-NP, corresponding to Au crystal on MOF framework." A reference is needed.

Reply: According to the reviewer's suggestion, we have added a reference.

Thank you!

Reviewer #3:

Comments:

The authors performed an experimental study on Visible-light-switched electron transfer over single porphyrin-metal atom center for the highly selective electrochemical reduction of CO_2 after carefully go through the whole manuscript, some comments for this work are as follows:

Reply: Thank you for realizing the importance and novelty of our work. We are extremely grateful for this reviewer's many constructive comments. To address these issues has helped to improve our manuscript quality. Therefore, we are delighted to make these wonderful revisions.

1. There are some literature reported electroreduction of CO_2 via porphyrin complex. it will be better if the authors can make a comparison (in a Table) of the performance (e.g., overpotential, faradaic efficiency, TOF, durability, stability, etc) between the current system and those from literature for the electroreduction of CO_2 . This will highlight the present study and also provide useful information for the community of CO_2 reduction, and further promote the progress of this field.

Reply: Thank you for your professional advice. We have added the performance comparison (Supplementary Table. 3) and the related discussion (Page 9, Line 20-22) in our revised manuscript.

2. In the first paragraph of introduction section, e.g., on line 32, electrocatalysis of CO₂, the authors should include and cite some latest references on CO₂ reduction and single atom catalysis.

Theor. Chem. Acc. 2018. 137. 98.

ACS Sustainable Chem. Eng. 2018, 6, 15494–15502.

J. Mater. Chem. A, 2019, 7, 3805–3814

J. Mater. Chem. A, 2019, DOI: 10.1039/C9TA01188A

<https://pubs.rsc.org/en/content/articlelanding/2019/TA/C9TA01188A#!divAbstract>

It will be better if the authors can compare the present study with that from J. Mater. Chem. A, 2019, DOI: 10.1039/C9TA01188A <https://pubs.rsc.org/en/content/articlelanding/2019/TA/C9TA01188A#!divAbstract>

This is the latest study on electroreduction of CO₂ by two dimensional poly-porphyrin monolayers. the unit cell is very close to the current porphyrin complex.

Reply: We thank the reviewer for recommending these latest computational work on CO₂ reduction and single atom catalysis, especially the one of two-dimensional poly-porphyrin monolayers, and we compared our research with them. Although the models vary, the results that overpotential of porphyrin-based single-atom Co catalysts is lower than the Cu catalysts, are qualitatively consistent in these researches, which indicates the catalytic properties are localized and influenced significantly by the metal center in porphyrin. According to the reviewer's suggestion, we have compared the latest study (Page 10, Line 10-13) in our revised manuscript.

3. On line 190, page 8, the authors used computational hydrogen electrode model (CHE) to study the pathway of CO₂ reduction. but on line 301, page 13, the authors used Gaussian 16 with PBE0+D3 functional to do calculations.

It should be pointed out that CHE model was proposed and used in the periodic systems, e.g., metal surfaces.

But Gaussian code usually used for cluster systems. The authors need to explain the more details about the calculations of CO₂ reduction and how to used CHE model with Gaussian code.

Reply: In our calculations, the computational hydrogen electrode model was used to relate the electrode potential to the Gibbs free energy change of a half electrochemical reaction, and the standard hydrogen electrode was used as reference potential electrode, where H⁺ + e⁻ pair is in equilibrium with gas-phase H₂ at 298 K, 0 V, pH=0 and 1 bar, and thus G(H⁺ + e⁻) = G(1/2 H₂). Although computational hydrogen electrode model was first proposed by Nørskov et al. (J. Phys. Chem. B 2004, 108, 17886) to study the electrochemical processes on metal surfaces with a slab approach, it has been successfully extended to other electrocatalysts such as metal nanoparticles based on a finite model, and Gaussian code has also been used to calculate the Gibbs free energies in many studies, which is particularly suitable to describe localized properties or catalytic reactions taking place at active sites [Chemical science, 8(1), 458-465, International Journal of Quantum Chemistry, 2016, 116(22), 1623-1640]. In fact, the single-atom catalyst, HNTM-Au-SA, used in our research is a series of porphyrinic metal-organic frameworks (MOFs) and the catalytic reactions are taking place at metal-coordinated porphyrin centers, where the electronic structures are supposed to be localized. Thus, we used Gaussian code and finite molecular model to study CO₂ reduction reaction mechanism. The Gibbs free energy change under the zero electrode potential for each reaction step was calculated by the equation:

$$\Delta G = \Delta E + \Delta ZPE + \Delta_{0 \rightarrow 298} H - T \Delta S,$$

in which the metal-coordinated porphyrins were fixed after optimization and thus zero-point vibrational energy (ZPE), thermal energy (H) and entropy (S) contributions of them were excluded from the calculations.

Revision made:

We add “The Gibbs free energy change under the zero electrode potential for each reaction step was calculated by the equation $\Delta G = \Delta E + \Delta ZPE + \Delta_{0 \rightarrow 298} H - T \Delta S$,” in

which the metal-coordinated porphyrins were fixed after optimization and thus zero-point vibrational energy (ZPE), thermal energy (H) and entropy (S) contributions of them were excluded from the calculations.” to the supporting information.

4. There are some problems in the reference citation
ref.22 < Adv. Mate.r> should be < Adv. Mater>
ref.2 < Chem. Soc. Rev. 40, 3703-3727 (2011)> should be < Chem. Soc. Rev. 40, 3703-3727 (2011)>
There should be only a space between the journal name and the number of volume.
ref.1 please double check the name of the journal <Earth System Science Data>, make sure it is abbreviation.

Reply: We have corrected this mistake in the revised manuscript. Thank you!

Reviewers' comments:

Reviewer #1 (Remarks to the Author):

I believe all comments aroused by the reviewer have been tackled correctly and the article is ready for publication.

Reviewer #2 (Remarks to the Author):

The authors have addressed many of the comments provided during the first round of reviews. These efforts are commended. Some of the new text and analyses added (or absence of these new analyses) raise further questions which I outline below.

- Regarding the "light filed effect": I was not asking for what it means in terms of the mechanism. Instead I am confused by the term "filed". Shouldn't it be "field" instead?

Also, what does "external filed input" mean? What does "plasma filed" mean?

The authors should pay attention throughout the manuscript, they seem to confuse the terms "filed" and "field". In many instances though, even if the term "field" is used, I am not sure this term is appropriately used...

- "Here, we recognize the effect and the catalyst as "light filed effect" and "photo-coupled electrocatalyst", respectively." I don't understand what the respectively term refer to. Overall, the new section added to the introduction reads very poorly.

- "Oppositely, whether light filed can interfere small molecule activation (such as CO₂) on electrocatalyst may be ambiguous but meaningful point. In this work, we try to seek novel electrocatalysts that can couple with light, and assist us to further improve activity as well as understand the coupling effect." I don't understand this sentence.

- "the most excellent electrocatalytic ability": what does this mean? The authors should provide a more rigorous and technical phrasing.

- Given the newly added Figure 5d, how can one say that the light has a particular impact on the process? This might only be the case for HNTM-Cu-NP and SA. It seems to be within the error range for the other samples.

- The authors performed tests under N₂ to check the origin of CO. The recommended approach in the field, especially for C-containing materials is to use ¹³CO₂. I would expect this test to be performed given the quality of work the authors are targeting.

- Figure S3d: why is Au mapping not overlapping the hollow fiber but taking the whole space?

Reviewer #2:

The authors have addressed many of the comments provided during the first round of reviews. These efforts are commended. Some of the new text and analyses added (or absence of these new analyses) raise further questions which I outline below.

Reply: Thank you very much for recognizing our efforts. We are extremely grateful for this reviewer's many constructive comments. To address these issues has helped to improve our manuscript quality. Therefore, we are delighted to make these wonderful revisions.

- Regarding the "light filed effect": I was not asking for what it means in terms of the mechanism. Instead I am confused by the term "filed". Shouldn't it be "field" instead? Also, what does "external filed input" mean? What does "plasma filed" mean? The authors should pay attention throughout the manuscript, they seem to confuse the terms "filed" and "field". In many instances though, even if the term "field" is used, I am not sure this term is appropriately used...

Reply: Thank you once again for pointing out this mistake. We have corrected the wrong spelling and checked the term throughout the manuscript. Thank you!

- "Here, we recognize the effect and the catalyst as "light filed effect" and "photo-coupled electrocatalyst", respectively." I don't understand what the respectively term refer to. Overall, the new section added to the introduction reads very poorly.

Reply: Thank you for pointing this out. We have corrected this sentence in the introduction (Page 2, Line 16-18).

- "Oppositely, whether light filed can interfere small molecule activation (such as CO₂) on electrocatalyst may be ambiguous but meaningful point. In this work, we try to seek novel electrocatalysts that can couple with light, and assist us to further improve activity as well as understand the coupling effect." I don't understand this sentence.

Reply: We have corrected this sentence and made it more clearly according to the reviewer's suggestion (Page 2, Line 10-13). Thank you!

- “the most excellent electrocatalytic ability”: what does this mean? The authors should provide a more rigorous and technical phrasing.

Reply: According to the reviewer’s suggestion, we have changed “the most excellent electrocatalytic ability” to “the best catalytic activity” (Page 3, Line 10) in our revised manuscript. Thanks!

- Given the newly added Figure 5d, how can one say that the light has a particular impact on the process? This might only be the case for HNTM-Cu-NP and SA. It seems to be within the error range for the other samples.

Reply: Thanks for your professional suggestion. These tests were carefully repeated to reduce error, as shown in Fig 3d, 3c and 4a. Fig 3d clearly indicates that light irradiation can promote CO₂ reduction at a lower overpotential on HNTM-Au-SA. On the contrary, HNTM-Co-SA shows a negligible change of TOF curve, corresponding to its energy gap of 0 eV (Fig 4c). Compared with HNTM-M-SA, HNTM-M-NP usually shows less improvement on electrocatalytic activity under visible light, indicating nanoparticle is difficult to be affected by light irradiation.

Fig. 3d TOF curves of HNTM-Au-SA under visible light (red line)/dark (black line).

- The authors performed tests under N₂ to check the origin of CO. The recommended approach in the field, especially for C-containing materials is to use ¹³CO₂. I would expect this test to be performed given the quality of work the authors are targeting.

Reply: To prove the origin of the products, we carried out ^{13}C -labeled isotopic experiments by using $^{13}\text{CO}_2$ as carbon source, and the generated CO and HCOOH was analyzed by GC-MS and ^1H NMR, respectively. As shown in Fig. S13a,b, only ^{13}CO ($m/z=29$) was detected on HNTM-Au-SA and HNTM-Co-SA, which is different from the ^{12}CO ($m/z=28$) when using $^{12}\text{CO}_2$ as carbon source. In addition, the peak assigned to $\text{H}^{12}\text{COO}^-$ is observed at 8.3 ppm (Fig. S13c). However, the ^1H NMR spectrum of the electrolyte exhibits a doublet after $^{13}\text{CO}_2$ electrocatalysis, which is attributed to the methine proton of $\text{H}^{13}\text{COO}^-$. These results prove that both CO and HCOOH are originated from CO_2 reduction, and not from the organic residue in the electrolyte or MOF material. We have added related description in our revised manuscript (see Page 7 and 8).

a

b

c

Supplementary Figure 13. Isotope labeling study. Mass spectra of ^{12}CO ($m/z=28$) formed on **(a)** HNTM-Au-SA and **(b)** HNTM-Co-SA. Insets show mass spectra of ^{13}CO ($m/z=29$) when $^{13}\text{CO}_2$ is used. **c**, ^1H NMR spectra of the electrolyte after $^{12}\text{CO}_2$ (blue spectrum) and $^{13}\text{CO}_2$ (red spectrum) electrolysis.

- Figure S3d: why is Au mapping not overlapping the hollow fiber but taking the whole space?

Reply: Thank you very much for pointing this out. According to your knowledge, the TEM sample holder of JEM-2010F FEI contains trace Au element that may increase noise signal and disturb element imaging. In addition, low Au loading leads to weak Au element signal, thus decreasing the image contrast.

To clearly observe Au element distribution, we measured EDS elemental mapping of HNTM-Au-SA once again. The mapping is shown below.

Supplementary Figure 3d EDS elemental mapping of HNTM-Au-SA.

REVIEWERS' COMMENTS:

Reviewer #2 (Remarks to the Author):

The authors have adequately address the last comment. Thank you.